# Antiproliferative Illudalane Sesquiterpenes from the Marine Sediment Ascomycete *Aspergillus oryzae*

**DOI:** 10.3390/md19060333

**Published:** 2021-06-10

**Authors:** Raha Orfali, Shagufta Perveen, Muhammad Farooq Khan, Atallah F. Ahmed, Mohammad A. Wadaan, Areej Mohammad Al-Taweel, Ali S. Alqahtani, Fahd A. Nasr, Sobia Tabassum, Paolo Luciano, Giuseppina Chianese, Jyh-Horng Sheu, Orazio Taglialatela-Scafati

**Affiliations:** 1Department of Pharmacognosy, College of Pharmacy, King Saud University, P.O. Box 2457, Riyadh 11451, Saudi Arabia; afahmed@ksu.edu.sa (A.F.A.); amaltaweel@ksu.edu.sa (A.M.A.-T.); alalqahtani@ksu.edu.sa (A.S.A.); 2Bio-Products Research Chair, Department of Zoology, College of Science, King Saud University, P.O. Box 2455, Riyadh 11451, Saudi Arabia; fmuhammad@ksu.edu.sa (M.F.K.); wadaan@ksu.edu.sa (M.A.W.); 3Department of Pharmacognosy, Faculty of Pharmacy, Mansoura University, Mansoura 35516, Egypt; 4Medicinal, Aromatic and Poisonous Plants Research Center, College of Pharmacy, King Saud University, P.O. Box 2455, Riyadh 11451, Saudi Arabia; fnasr@ksu.edu.sa; 5Interdisciplinary Research Centre in Biomedical Materials (IRCBM), Lahore Campus, COMSATS University Islamabad, Islamabad 44000, Pakistan; sobiatabassum@cuilahore.edu.pk; 6Department of Pharmacy, School of Medicine and Surgery, University of Naples Federico II, Via Montesano 49, 80131 Naples, Italy; pluciano@unina.it (P.L.); g.chianese@unina.it (G.C.); 7Department of Marine Biotechnology and Resources, National Sun Yat-sen University, Kaohsiung 80424, Taiwan; sheu@mail.nsysu.edu.tw; 8Department of Medical Research, China Medical University Hospital, China Medical University, Taichung 40447, Taiwan

**Keywords:** *Aspergillus oryzae*, marine fungus, illudalane sesquiterpenes, antiproliferative activity, zebrafish toxicity

## Abstract

The new asperorlactone (**1**), along with the known illudalane sesquiterpene echinolactone D (**2**), two known pyrones, 4-(hydroxymethyl)-5-hydroxy-2*H*-pyran-2-one (**3**) and its acetate **4**, and 4-hydroxybenzaldehyde (**5**), were isolated from a culture of *Aspergillus oryzae*, collected from Red Sea marine sediments. The structure of asperorlactone (**1**) was elucidated by HR-ESIMS, 1D, and 2D NMR, and a comparison between experimental and DFT calculated electronic circular dichroism (ECD) spectra. This is the first report of illudalane sesquiterpenoids from *Aspergillus* fungi and, more in general, from ascomycetes. Asperorlactone (**1**) exhibited antiproliferative activity against human lung, liver, and breast carcinoma cell lines, with IC_50_ values < 100 µM. All the isolated compounds were also evaluated for their toxicity using the zebrafish embryo model.

## 1. Introduction

The marine environment is an unsurpassed casket of chemodiversity and a prolific source of biologically active compounds with potential medicinal applications [1]. In the last 50 years, scientists all over the world have dedicated their efforts to uncover the potential of marine metabolites, succeeding in the isolation of thousands of natural products with peculiar architectures and interesting bioactivities [2]. As per April 2021, the marine pharmacology arsenal [3] includes 15 approved drugs (mainly for cancer treatment), seven compounds in phase I, 12 compounds in phase II, and 5 compounds in phase III clinical trials, the latter including plitidepsin, recently proposed for COVID-19 symptomatic treatment [4,5]. Sponges and tunicates are undoubtedly the most intensely studied marine organisms, but it is now clear that marine microbes, isolated from sediments or from symbiotic plants or invertebrates, constitute a rich source for secondary metabolites. In this context, although terrestrial fungi are more explored, in comparison to their marine counterparts, a surprising number of structurally unique compounds have been isolated from fungi living in marine habitats [6]. The peculiar properties of the marine environment with regard to nutrients, temperature, and competition, are likely crucial factors in improving the ability of marine fungi to elaborate compounds with promising bioactivities, especially suited for antibiotic and anticancer applications [7]. 

The genus *Aspergillus* is one of the most abundant among marine ascomycetes, characterized by high salt tolerance, fast growth rate, and the capacity to adapt to diverse habitats. Marine *Aspergillus* fungi produce a wide range of secondary metabolites belonging to different classes and are endowed with a broad array of biological activities of industrial and pharmaceutical interest [8,9]. As a part of our ongoing research activity aimed at the isolation of bioactive compounds from terrestrial and marine fungi [10,11,12,13], *A. oryzae* samples obtained from the sediments of the Red Sea, along the coasts of Saudi Arabia, were chemically investigated. This study resulted in the isolation of two illudalane sesquiterpenes, the new asperorlactone (**1**) and the known echinolactone D (**2**), along with two rare pyrone derivatives (**3**–**4**) and 4-hydroxybenzaldehyde (**5**). These compounds were evaluated for their antiproliferative activity against three human carcinomas (lung, liver, and breast) cell lines. Chemotherapeutics used in cancer treatment are often characterized by marked toxic effects on normal cells. Less than 2% of compounds emerging from in vitro drug screening could enter clinical trials since the majority of the newly developed leads fail in preclinical testing due to their toxicity in experimental animal models [14]. In vitro drug screening methods are prevalently used in order to characterize the bioactivity and the toxicity of new compounds, while testing their safety in suitable animal models prior to clinical trials would save time and money. Since a high throughput screening approach is not feasible in higher animals, zebrafish constitute a valid option for preclinical testing and have shown quite reproducible results. It is predicted that, following further development of technologies, zebrafish will play a key role in speeding up the emergence of precision medicine [15,16]. On these bases, the isolated compounds **1**–**5** have been tested for their zebrafish animal toxicity, and results are reported herein.

## 2. Results and Discussion

### 2.1. Extraction and Structural Identification

*A. oryzae* samples were isolated from the Red Sea sediment collected at a depth of—50 m off Jeddah, Saudi Arabia. Fermentation of the fungus on solid rice medium and extraction with EtOAc afforded a brownish residue, which was chromatographed by using silica gel and RP-18 column chromatography, affording one new (**1**) and four known (**2**–**5**) compounds (Figure 1). 

The known compounds were identified as echinolactone D (**2**) [17], 4-(hydroxymethyl)-5-hydroxy-2*H*-pyran-2-one (**3**) [18], (5-hydroxy-2-oxo-2H-pyran-4-yl)methyl acetate (**4**) [16], and 4-hydroxybenzaldehyde (**5**), by a comparison of their spectroscopic data with those reported in the literature. The illudalane echinolactone D (**2**) had been isolated before from mycelia of the fungus *Echinodontium japonicum* [17] and from the wood decomposing fungus *Granulobasidium vellereum* [19], and therefore, this is its first report from a marine source. Compound **3** is a rare isomeric analog of kojic acid for which only three reports were present in the literature, all from *Aspergillus* fungi (marine *A. flavus* [18], terrestrial *A. niger* [20], and freshwater *A. austroafricanus* [21]). 

Asperorlactone (**1**) was isolated as a colorless oil with molecular formula C_15_H_18_O_3_, determined by HR-ESIMS. The ^13^C NMR and DEPT spectra of **1** (Table 1) showed the presence of one lactone carbonyl (δ_C_ 166.5), one aromatic methine (δ_C_ 123.1), one oxymethine (δ_C_ 60.8), one sp^3^ (δ_C_ 39.3) and five sp^2^ (δ_C_ 122.7, 132.2, 136.2, 144.3, 150.1) nonprotonated carbons, three sp^3^ methylenes (δ_C_ 46.5, 47.1, 72.7, the latter O-bearing), and three methyl groups (δ_C_ 13.1, 27.5 × 2). On the basis of these data, the seven unsaturation equivalents required by the molecular formula of **1** could be accommodated by the presence of a benzene ring and of two additional rings, including a lactone.

The ^1^H NMR spectrum of **1** (Table 1) showed only one aromatic methine signal (δ_H_ 7.75), two methyl singlets (δ_H_ 1.19 × 2), an arylmethyl at δ_H_ 2.37 (s, 3H), and two pairs of methylenes around δ_H_ 2.80–2.83. The single-spin system of **1** included a diastereoptopic hydroxymethylene (δ_H_ 4.52 and 4.63), and an oxymethine signal (δ_H_ 4.95). Having associated all these proton signals to those of the directly linked carbons with the 2D NMR HSQC experiment, the illudalane type sesquiterpenes skeleton of compound **1** could be established by following the correlation network of the 2D NMR HMBC spectrum (Figure 2) (see Appendix A). 

In particular, correlations of H-9 with C-8 and the lactone C-7, of H-5 with C-4 and C-8, and of H_2_-6 with C-7 defined the structure of the hydroxyisochroman-1-one moiety of asperorlactone. The structure of the condensed dimethylated five-membered ring was deduced from the correlation of H_3_-14/H_3_-15 with C-1, C-11, and C-12 and of H_2_-1 and H_2_-11 with C-2 and C-10. Finally, the remaining methyl group was attached at C-3, following its correlations with C-2, C-3, and C-4.

Asperorlactone (**1**) is an optically active compound (${[\alpha ]}_{D}^{23}$ − 19.8) with a single stereocenter (C-5). We first tried to define the absolute configuration of **1** by employing the modified Mosher’s method [22]; however, all the attempts to obtain the formation of the MTPA esters from the corresponding chloride failed. Most likely, the reaction of the bulky MTPA group with the hydroxyl group at C-5 was hindered by the methyl group present on the condensed aromatic ring. Therefore, we decided to rely on computational calculations, reasoning that comparison between experimental and quantum mechanically calculated ECD spectrum could provide an unambiguous indication on the absolute configuration of **1**.

The structure of **1** was subjected to a geometry and energy optimization using DFT with the mPW1PW91/6-311+G (d,p) functional and basis set combination using the Gaussian 09 software. The reasonably populated conformations, their relative energy, and the equilibrium room-temperature Boltzmann populations are reported in Figure 3. The two major conformers **1a** and **1b**, accounting for 99.5% of the total population, differ almost exclusively for the pseudorotation of the five-membered ring.

TDDFT calculations were run using the functional CAM-B3LYP and the basis sets 6-31G (d,p) including at least 30 excited states in all cases, and using IEF-PCM for MeOH. The rotatory strength values were summed after a Boltzmann statistical weighting, and Δε values were calculated by forming sums of Gaussian functions centered at the wavelengths of the respective electronic transitions and multiplied by the corresponding rotatory strengths. Thus, the ECD spectra for *R*-**1** and *S*-**1** were obtained (Figure 4). The extensive similarity of the first with the experimental ECD spectrum allowed us a confident assignment of the absolute configuration of asperorlactone as 5*R*.

The isolation of asperorlactone (**1**), as well as of the related echinolactone D (**2**), is of great relevance because, to our knowledge, this is the first report of illudalane-type sesquiterpenes from an ascomycete (*A. oryzae*), since this class of metabolites has, until now, been found exclusively in basidiomycetes.

It has been reported that illudalanes derive biosynthetically from a humulene precursor that, upon cyclization, would generate a protoilludane that finally would rearrange to form the illudalane derivative [23]. In the light of this hypothesis, a possible biosynthesis of asperorlactone is reported in Figure 5, where illudol [24] is a key intermediate that could afford **1** by dehydration, oxidation, and four-membered ring opening.

### 2.2. Pharmacological and Toxicological Evaluation of the Isolated Compounds

Illudalane sesquiterpenes, obtained from different sources, have been reported to possess several biological properties, with a special focus on anticancer activities [25,26]. This prompted us to evaluate the antiproliferative activity of compounds **1**–**5** against three human cancer cell lines, namely, lung carcinoma (A549), liver carcinoma (HepG2), and breast carcinoma (MCF7). The obtained IC_50_ values are presented in Table 2. Compounds **1**–**5** generally showed moderate antiproliferative activity against all tested cell lines, invariably with higher potency against lung carcinoma, as compared to liver or breast carcinoma. As shown in Table 1, asperorlactone (**1**) and compound **2** were the most potent compounds against three cancer cell lines, with the single exception of the activity of **5** against the MCF-7 cell line. Additionally, interesting is the comparison between compounds **3** and **4**, evidencing that acetylation improves the activity against A549 and HepG2.

As anticipated, screening in zebrafish embryos provides an excellent environment in which the toxicity of a compound on noncancer cells and systems could be predicted. Thus, the zebrafish embryos model was used to evaluate the animal toxicity of compounds **1**–**5** (Figure 6). The LC_50_ values (the concentration required to kill 50% of embryos) of all the tested compounds were higher than the 1 mg/mL range, indicating their safety on noncancer cells and selectivity indices >50. The zebrafish embryos that were treated with compounds **2**, **3**, and **5** did not exhibit any observable toxicity, and they developed normally up to 3 days post treatment. On the other hand, compound **4**, for which the initial development and growth of zebrafish embryos was normal, induced the death of 100% of treated embryos after 24 h post treatment. The zebrafish embryos that were treated with more than 200 µM compound **1** (higher than IC_50_) developed normally; however, the embryos exhibited cardiac toxicity (cardiac edema and cardiac hypertrophy, black arrow in Figure 6) after 2 days post treatment.

## 3. Materials and Methods

### 3.1. Fungal Material

*Aspergillus oryzae* was isolated from the marine sediment collected at −50 m off Jeddah, Red Sea, Saudi Arabia, in October 2018. The fungal identification was conducted by DNA amplification and sequencing of the internal transcribed spacer region (GenBank accession No. MH608347) followed by a subsequent BLAST search in NCBI according to the protocol described before [27]. The specimen of the fungal strain was deep-frozen and deposited at the authors’ lab (R.O., S.P.).

### 3.2. Fermentation, Extraction, and Isolation

*A. oryzae* was cultivated in 20 Erlenmeyer flasks on solid rice medium containing (100 g rice, 3.5 g sea salt, and 110 mL of demineralized water). After autoclaving at 121 °C for 20 min and then cooling to room temperature, each flask was inoculated and then incubated at 20 °C under static conditions. After four weeks, 500 mL EtOAc was added to each flask to stop the fermentation and extract. The total extract was collected after the flasks had been shaken at 150 rpm for 8 h on a laboratory shaker. The obtained crude extract after evaporation of the EtOAc (7 g) was then partitioned between *n*-hexane and MeOH. The polar phase was then subjected to Sephadex LH-20 column (100 × 2.5 cm) using 100% methanol as an eluting solvent. Similar fractions were combined with each other according to TLC readings and further purified by semipreparative HPLC using gradient system MeOH-H_2_O from 40:60 to 70:30 in 30 min to afford **1** (3.5 mg), **2** (5.0 mg), **3** (7.0 mg), **4** (5.0 mg), and **5** (2.6 mg).

*Asperorlactone (**1**).* Colorless oil; ${[\alpha ]}_{D}^{23}$−19.8 (*c* 0.6, MeOH); UV λ_max_ (MeOH) nm (log ε): 230 (4.6); ECD λ_max_ (MeOH) nm (Δε): 238 (+ 9.6), 265 (−4.8); ^1^H NMR (700 MHz, CD_3_OD), and ^13^C NMR (175 MHz, CD_3_OD); see Table 1; ESIMS *m/z* 269.1150 [M + Na]^+^ (calc. for C_15_H_18_ O_3_ Na *m/z* 269.1154).

### 3.3. Computational Calculations

A preliminary conformational search on each stereoisomer was performed by Simulated Annealing in the INSIGHT II package. The MeOH solution phases were mimicked through the value of the corresponding dielectric constant. Using the steepest descent followed by quasi-Newton–Raphson method (VA09A) the conformational energy was minimized. Restrained simulations were carried out for 500 ps using the CVFF force field as implemented in Discovery software (Version 4.0 Accelrys, San Diego, USA). The simulation started at 1000 K, and then the temperature was decreased stepwise to 300 K. The final step was again the energy minimization, performed in order to refine the conformers obtained, using the steepest descent and the quasi-Newton–Raphson (VA09A) algorithms successively. Both dynamic and mechanic calculations were carried out by using 1 (kcal/mol)/A˚ 2 flat well distance restraints. In total, 100 structures were generated. TDDFT calculations were run using the functional CAM-B3LYP and the basis sets 6-31G (d,p) including at least 30 excited states in all cases, and using IEF-PCM for MeOH. The rotatory strength values were summed after a Boltzmann statistical weighting, and Δε values were calculated by forming sums of Gaussian functions centered at the wavelengths of the respective electronic transitions and multiplied by the corresponding rotatory strengths. The ECD spectra that were obtained were UV-corrected and compared with the experimental ones.

### 3.4. In Vitro Antiproliferative Activity

The antiproliferative activity of compounds was measured using MTT (3-(4, 5-dimethylthiazol-2-yl)-2,5-diphenyltetrazolium bromide) assay. A549 (lung), HepG2 (liver), and MCF-7 (breast) cancer cells were purchased from the American Type Cell Collection (ATCC, Manassas, VA, USA). The cells were seeded at 5 × 10^4^ cells/well (in 100 μL of DMEM) in 96-well microplates. After 24 h incubation at 37 °C, serial dilution (15–250 µM) of each compound was added and incubated for 48 h. Thereafter, 10 μL of the MTT solution (5 mg/mL) was added to each well. After 4 h incubation with the MTT solution, a volume of 100 μL of acidified isopropanol was added to solubilize the formazan product and incubated on a shaker for a further 10 min. Reduced MTT was assayed at 570 nm using a microplate reader (BioTek, Winoosky, VT, USA). Control groups received the same amount of DMSO (0.1%), untreated cells were used as a negative control, whereas cells treated with doxorubicin were used as a positive control. The IC_50_ (concentration that caused more than 50% inhibition of proliferation) was calculated from a dose-dependent curve. The cell viability percent was calculated = mean absorbance of treated sample/mean absorbance of control ×100.

### 3.5. Zebrafish Toxicity Screening

Wild-type zebrafish strain AB/Tuebingen TAB-14(AB/TuebingenTAB-14) (Catalog ID: ZL1438) were obtained from zebrafish international resource center and grown in an animal facility at the Department of Zoology, King Saud University, Riyadh, Kingdom of Saudi Arabia. The fish were maintained and bred following guidelines of the Institutional Animal Care and Use Committee (ICUAC) and zebrafish book. The fertilized embryos were obtained by natural pairwise breeding of adult fish. The fertilized embryos were sorted, dead embryos were removed, and synchronous stage embryos were used for screening.

A stock solution of 25 mM was made by dissolving the compounds in molecular biology grade DMSO (Sigma Aldrich, St. Louis, MI, USA). Zebrafish embryos were exposed to serial dilution (1, 5, 15, 45, 150, and 300 µM) of each compound. The embryos were remained exposed to the compounds for 3 days, and the embryos medium containing the compounds were changed every day. The response of the embryos toward mortality, and embryonic toxicity (teratogenicity) was monitored once after 12 h and then after every 24 h until the end of the experiment. The experiment was repeated at least three times (triplicate biological repeats) by using a new batch of embryos every time. LC_50_ for zebrafish embryonic toxicity was calculated by using an updated Probit analysis by Finney method [28].

## 4. Conclusions

Samples of *A. oryzae*, obtained from the Red Sea sediments collected off Jeddah, Saudi Arabia, afforded the first illudalane sesquiterpenoids isolated from an ascomycete, including the new asperorlactone (**1**), characterized by hydroxylation of the lactone ring. Elucidation of the structure and absolute configuration of this compound needed application of ESIMS and NMR spectral analyses and computational calculations of ECD spectra. The isolated compounds showed moderate antiproliferative activity against three human carcinoma cell lines (lung, liver, and breast). The zebrafish embryo model was used to evaluate the animal toxicity of compounds and selected echinolactone D (**2**) and the rare pyrone **3** as targets worthy of further investigation.

## Figures and Tables

**Figure 1 marinedrugs-19-00333-f001:**
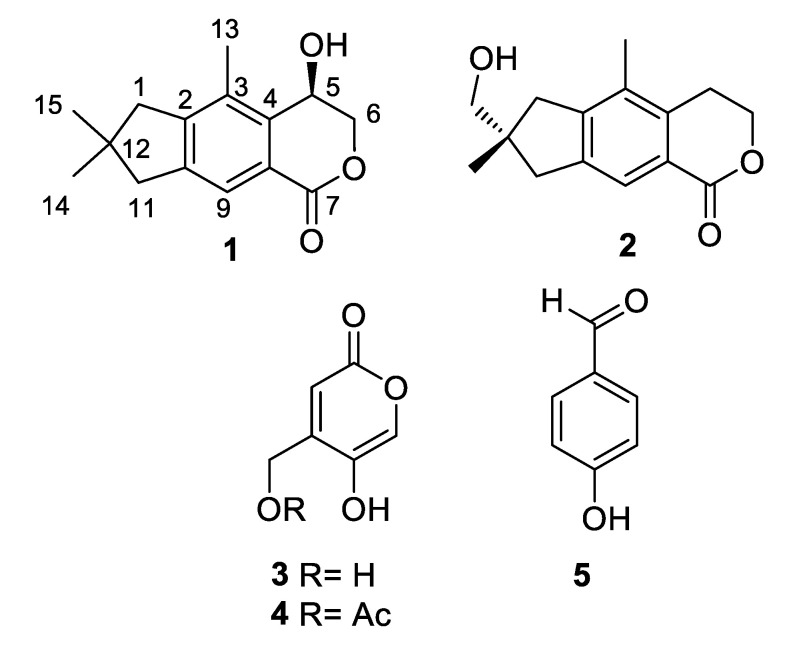
Chemical structures of metabolites isolated from *A. oryzae*.

**Figure 2 marinedrugs-19-00333-f002:**
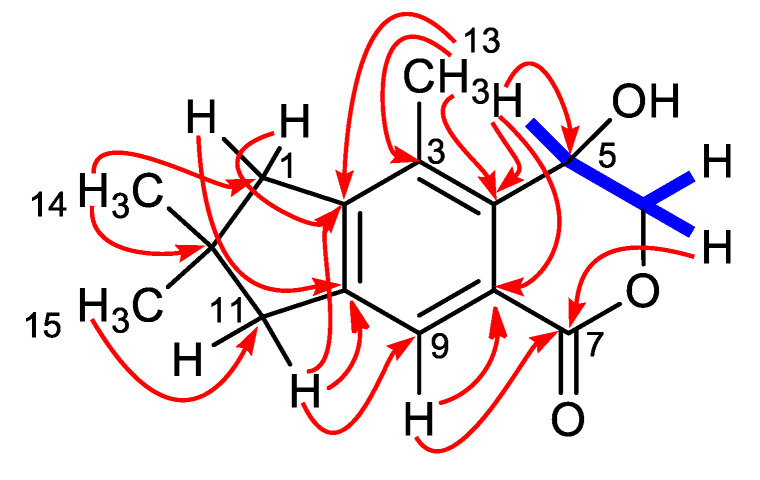
COSY (blue) and key HMBC (red arrows) correlations for compound **1**.

**Figure 3 marinedrugs-19-00333-f003:**
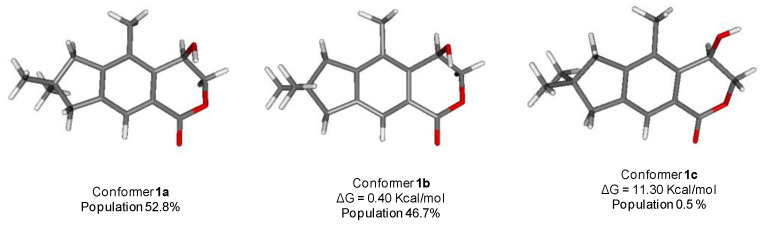
The reasonably populated conformers **1a**–**1c** of **1** and their calculated Boltzmann population.

**Figure 4 marinedrugs-19-00333-f004:**
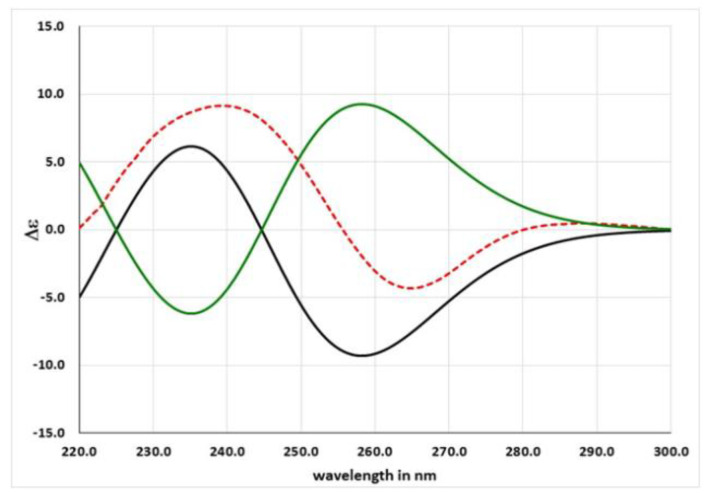
Experimental ECD curve of asperorlactone (red) and calculated ECD curves for *R*-**1** (black) and *S*-**1** (green).

**Figure 5 marinedrugs-19-00333-f005:**
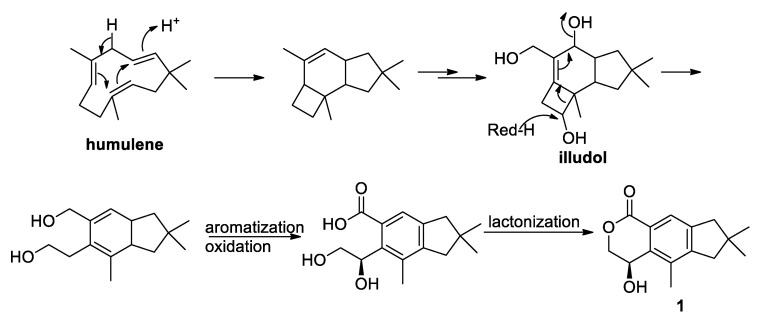
Postulated biosynthesis of asperorlactone (**1**).

**Figure 6 marinedrugs-19-00333-f006:**
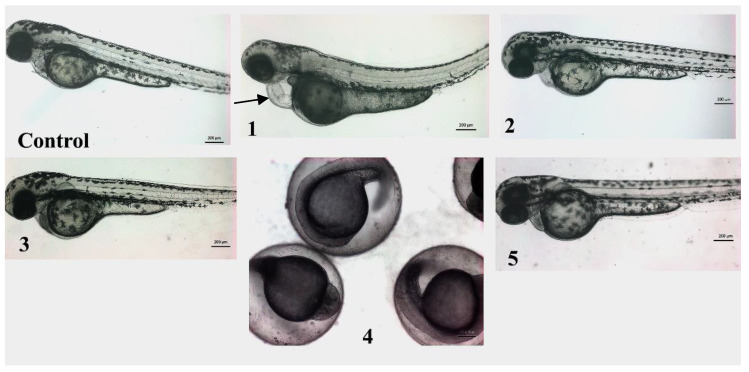
In vivo screening of compounds **1**–**5** in zebrafish embryos. Representative micrograph of embryos at 3 days poster fertilization, which were treated with compounds **1**–**5**. The zebrafish embryos treated with compounds **2**, **3**, and **5** developed normally, and there were no obvious differences in morphology and growth between control and treated embryos. The zebrafish embryos treated with >200 µM of **1**, however, had cardiac edema and cardiac hypertrophy (black arrow). The zebrafish embryos treated with compound **4** developed normally but were found dead on day 2. All the images are in same magnification, scale is 200 µm.

**Table 1 marinedrugs-19-00333-t001:** ^1^H (700 MHz) and ^13^C (175 MHz) NMR data for asperorlactone (**1**) in CD_3_OD.

Positions	δ_H_ (Mult., *J* in Hz)	δ_C_, Type
1a	2.83 (overlapped)	46.5, CH_2_
1b	2.81 (d, 17.5)	
2	-	150.1, C
3	-	132.2, C
4	-	136.2, C
5	4.95 (dd, 1.0, 2.1)	60.8, CH
6	4.63 (dd, 1.0, 12.0)	72.7, CH_2_
	4.52 (dd, 2.1, 12.0)	
7	-	166.5, C
8	-	122.7, C
9	7.75 (s)	123.1, CH
10	-	144.3, C
11	2.82 (overlapped)	47.1, CH_2_
12	-	39.3, C
13	2.37 (s)	13.1, CH_3_
14	1.19 (s)	27.5, CH_3_
15	1.19 (s)	27.5, CH_3_

**Table 2 marinedrugs-19-00333-t002:** Antiproliferative activity of compounds **1**–**5** against three human cancer cell lines.

Compound	IC_50_ (µM) for the Different Carcinoma Cell Lines		
	A549 (Lung)	HepG2 (Liver)	MCF-7 (Breast)
**1**	72.7 ± 1.1	86.6 ± 3.2	106.5 ± 4.2
**2**	55.7 ± 2.5	148.4 ± 5.6	128.0 ± 2.8
**3**	208.5 ± 6.8	220.4 ± 3.6	225.4 ± 5.1
**4**	89.4 ± 2.3	126.8 ± 6.4	170.7 ± 4.5
**5**	97.5 ± 2.6	242.6 ± 6.4	158.2 ± 5.5
Doxorubicin	2.1 ± 0.08	2.2 ± 0.15	1.9 ± 0.05

## Data Availability

The data presented in this study are available in the article and Appendix A.

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
