# Peer review of "Antiproliferative Illudalane Sesquiterpenes from the Marine Sediment Ascomycete Aspergillus oryzae"

_marinedrugs, 2021, doi:10.3390/md19060333_

Round 1

Reviewer 1 Report

This paper is focused on the isolation, purification and identification of one Illudalane-type sesquiterpenoid, isolated from the marine sediment ascomycete Aspergillus oryzae, along with four known derivatives. The new compound, named asperorlactone,is well described through detailed analysis of mass, 1D and 2D NMR data and computational calculations of ECD spectra for establishing its stereostructure. A biosynthesis schema is also proposed. Asperorlactone and its related derivative echinolactone D are the first illudalane-type sesquiterpenoids reported from an ascomycete, previously exclusively found in basidiomycetes. The antiproliferative activity of all isolated compounds was evaluated against three cancer cell lines (lung carcinoma A-549, liver carcinoma Hep-G2, and breast carcinoma MCF-7). Their toxicity was evaluated in the zebrafish animal toxicity assay.

The paper sounds good and provides the appropriate information for identification of the new illudalane sesquiterpenoid, including its absolute configuration at C-5. It showed good antiproliferative activity against the three cancer cell lines with IC50values < 30 μg/mL. After 2 days of treatment with > 50 μg/mL of asperorlactone, zebrafish embryos revealed cardiac oedema and cardiac hypertrophy.  

My main remark concerns the unit of the antiproliferative activity and toxicity results. It is surprising that compounds are tested at different concentration expressed in “μM”in the methods part and that results (IC50values) in Table 2 and in Figure 6, as well as in the body text, are presented in “μg/mL”. Please, check and modify if necessary.

Please, add these minor corrections:

- Page 2, Line 84: (5-hydroxy-2-oxo-2H-pyran-4-yl)methyl acetate (4)

- Page 3, line 93: The molecular formula is C15H18O3and not C15H20O3. Please, correct.

- Page 4: In figure 2, key HMBC correlations from H2-1 with C-2 and C-10 are lacking. Please, add them to be in accordance with the body text.

Author Response

My main remark concerns the unit of the antiproliferative activity and toxicity results. It is surprising that compounds are tested at different concentration expressed in “μM”in the methods part and that results (IC50values) in Table 2 and in Figure 6, as well as in the body text, are presented in “μg/mL”. Please, check and modify if necessary.

Answer:  As suggested, the unit of the antiproliferative activity and toxicity results were modified to molarity throughout the manuscript.

Please, add these minor corrections:

- Page 2, Line 84: (5-hydroxy-2-oxo-2H-pyran-4-yl)methyl acetate (4)

- Page 3, line 93: The molecular formula is C15H18O3and not C15H20O3. Please, correct.

Answer:  These minor corrections have been done

- Page 4: In figure 2, key HMBC correlations from H2-1 with C-2 and C-10 are lacking. Please, add them to be in accordance with the body text.

Answer:  These two correlations have been added in Figure 2.

Reviewer 2 Report

Recommendation: submit to a more appropriate journal due to lack of novelty and impact.

Comment to authors:

In this paper entitled “Antiproliferative Illudalane Sesquiterpenes from the Marine Sediment Ascomycete Aspergillus oryzae”, one new illudalane sesquiterpene, named asperorlactone (1), and four known compounds (2-5) were isolated from a culture of Aspergillus oryzae. The structure of 1 was determined by spectroscopic and spectrometric methods including 1D, 2D NMR, ECD and HRESIMS analyses. Antiproliferative activities against several human cancer cell lines as well as the toxicity using the zebrafish embryos were evaluated. Due to the lack of structural novelty and the weak activity profiles (> 100 μM as the best) of all the compounds described, I would recommend this manuscript to be published in a different journal, for example, Tetrahedron, Tetrahedron Letters, or Chemical and Pharmaceutical Bulletin.

General comments:

  1. The units of biological activity should be consistent through the whole manuscript. All the “μg/mL” in the manuscript should be corrected to “μM”. The 30 μg/mL IC50 for compound 1 was about 115.2 μM, the best activity value of compound 1 in Table 2 “17.9 μg/mL” is about 66.5 μM. The authors claimed all the isolated compounds showed good activity and in Lines 162 and 272, high potency. That is not accurate. In Line 237, the highest tested concentration for the antiproliferative activity was 40 μM, but all the IC50 values were higher, or much higher than that value. The authors should double check the experimental design.
  2. Please add proper references in the first paragraph in the introduction session, the first two sentences for example.
  3. Table 1 should be placed after Line 100. In addition, H2-1 and H2-11 were overlapped. The clear J coupling values were measured and reported for H2-1 but not for H2-11. Please double check the spectrum.
  4. The paragraph from Line 83 to 91 describing the known compounds should be place after Line 144 when the structure elucidation for the new compound 1 is done
  5. The biosynthesis was well proposed for the illudalane derivatives in the paper published in 1971 (Reference 21). The paragraph from Line 149 to 153 could be omitted as well as Figure 5.
  6. Image 4 should be consistent with the other ones in Figure 6.
  7. Please check the format through all of the references, especially the abbreviations, for example in Lines 316 and 317.

Specific/necessary corrections

  1. Line 21: The sentence should be rewritten as “The undescribed asperorlactone (1), along with one known illudalane sesquiterpene echinolactone D (2), two known pyrones…”.
  2. Line 24: The word “stereostructure” should be revised as “structure”.
  3. Lines 24 and 27: Please add “(1)” next to asperorlactone.
  4. Line 39: “7” should be corrected as “seven”
  5. Line 40: “5” should be corrected as “five”
  6. Line61: The word “lungs” should be revised as “lung”.
  7. Line 87: Please omit “only”.
  8. Lines 96 and 97: the orbital description sp2, sp3 should not be italic. The word “unprotonated” should be revised as “nonprotonated”. The sentence “two sp3 methylenes (δC5, 47.1), one sp3 oxymethylene (δC 72.7)” is misleading. It could be rewritten as “three sp3 methylenes (δC 46.5, 47.1, and 72.7 (O-bearing))”
  9. Line 102: “benzene-linking methyl” could be changed into “arylmethyl”. The proton signal (1.19 for H3-14 and H3-15) should also be stated here.
  10. Line 211: The UV and the CD wavelengths are different. Please compare with the sentence in Line 232 and double check to confirm.
  11. Line 226: The word “structures” should be corrected as “conformers”
  12. Line 260: The word “compound” should be corrected as “compounds”.
  13. Line 262: The words “hours” should be corrected as “h”.
  14. Line 272: “…showed good antiproliferative activity against…” should be corrected as “…showed weak antiproliferative activities against…”

Author Response

The units of biological activity should be consistent through the whole manuscript. All the “μg/mL” in the manuscript should be corrected to “μM”. The 30 μg/mL IC50 for compound 1 was about 115.2 μM, the best activity value of compound 1 in Table 2 “17.9 μg/mL” is about 66.5 μM. The authors claimed all the isolated compounds showed good activity and in Lines 162 and 272, high potency. That is not accurate. In Line 237, the highest tested concentration for the antiproliferative activity was 40 μM, but all the IC50 values were higher, or much higher than that value. The authors should double check the experimental design.

Answer: As suggested by the reviewer, the units of biological activity was corrected in terms of molarity unit through the whole manuscript. We agree with the reviewer and the indication of "good activity" has changed into "moderate activity". At line 237, the indication of 50 µg/mL has been changed into 200 µM.

Please add proper references in the first paragraph in the introduction session, the first two sentences for example.

Answer: References have been added for the first two sentences of Introduction. The numbers of following references have been shifted.

Table 1 should be placed after Line 100. In addition, H2-1 and H2-11 were overlapped. The clear J coupling values were measured and reported for H2-1 but not for H2-11. Please double check the spectrum.

Answer: The placement of Table 1 has been moved forward. H-11 is overlapped with H2-1a. This has been better indicated in the Table.

The paragraph from Line 83 to 91 describing the known compounds should be place after Line 144 when the structure elucidation for the new compound 1 is done

Answer: We prefer to report first the list of known compound, identified by comparison with the literature, and then describe in detail the structural elucidation of the new compound.

The biosynthesis was well proposed for the illudalane derivatives in the paper published in 1971 (Reference 21). The paragraph from Line 149 to 153 could be omitted as well as Figure 5.

Answer: The biosynthesis reported in the reference was generic for illudalane. We have adapted it to our compound and added further details, as requested by reviewer 3.

Image 4 should be consistent with the other ones in Figure 6.

Answer: Image 4 is not consistent with the other reported in Figure 6 because it is the only one reporting dead zebrafish.

Please check the format through all of the references, especially the abbreviations, for example in Lines 316 and 317.

Answer: We have checked the references and made a couple of corrections.

 Line 21: The sentence should be rewritten as “The undescribed asperorlactone (1), along with one known illudalane sesquiterpene echinolactone D (2), two known pyrones…”. Line 24: The word “stereostructure” should be revised as “structure”. Lines 24 and 27: Please add “(1)” next to asperorlactone.

Answer: These typographical corrections in the Abstract have been done.

Line 39: “7” should be corrected as “seven”; Line 40: “5” should be corrected as “five”; Line61: The word “lungs” should be revised as “lung”.

Answer: These typographical corrections in the Introduction have been done.

Line 87: Please omit “only”. Lines 96 and 97: the orbital description sp2, sp3 should not be italic. The word “unprotonated” should be revised as “nonprotonated”. The sentence “two sp3 methylenes (δC5, 47.1), one sp3 oxymethylene (δC 72.7)” is misleading. It could be rewritten as “three sp3 methylenes (δC 46.5, 47.1, and 72.7 (O-bearing))”. Line 102: “benzene-linking methyl” could be changed into “arylmethyl”. The proton signal (1.19 for H3-14 and H3-15) should also be stated here.

Answer: These corrections in the Results and Discussion have been done.

Line 211: The UV and the CD wavelengths are different. Please compare with the sentence in Line 232 and double check to confirm.

Answer: We have checked and confirm the values reported. However, it is not surprising that maxima in UV and CD do not exactly coincide.

Line 226: The word “structures” should be corrected as “conformers”. Line 260: The word “compound” should be corrected as “compounds”. Line 262: The words “hours” should be corrected as “h”.

Answer: These corrections in the Materials and methods have been done.

Line 272: “…showed good antiproliferative activity against…” should be corrected as “…showed weak antiproliferative activities against…”

Answer: We have changed "good" with "moderate"

Reviewer 3 Report

I have read the manuscript submission of Orfali et al. and hereby tender my findings.

Overall the manuscript is well written and logical in flow. There are only a handful of small grammatical errors, which I list below. In terms of scientific content, the manuscript is succinct but descriptive. I have a few suggested corrections:

  • My main suggestion is the naming of the new compound. Given its high structural similarity to the existing echinolactone family, especially echinolactone B, and that echinolactone D was also isolated, I think it is a stretch to give compound 1 a whole new name. I therefore request naming compound 1 as echinolactone E.
  • Line 21: The word novel is used far too often, and compound 1 is closely related to other known metabolites therefore replace “novel” with “new”. Novel should be restricted in use only to compounds with new carbon skeletons.
  • Line 24: Change to “… collected from Red Sea marine sediments.”
  • Line 29: Remove plural from zebrafish embryo. There is only one model here, not multiple.
  • Line 36: Remove several. Several implies only a few, and not the 100’s if not 1000’s of marine natural product chemists who have contributed over the years!
  • Line 57: Change “was” to “were” (there were multiple samples)
  • Line 98: Place spaces between “…27.5 x 2…” to make more clear.
  • Line 100: Remove one from end of sentence.
  • Line 102: Change to “…two pairs of…”
  • Line 108: Please add a few atom labels to the structure shown in figure 2 to assist the reader relate to the paragraph describing the 2D-NMR analysis.
  • Lines 110 – 115: Were there any HMBC correlations detected from H2-1? If so these should be added to figure 2 and described in the text as they would add value to the discussion.
  • Line 122: Change to “…experimental and the quantum mechanically…”
  • Line 128: Change to “… are shown in Figure 3.”
  • Line 147: Change to “… metabolites has, until now, been found…”
  • Line 154: I was confused by the proposed biosynthesis of 1 as I could not easily rationalize the opening of illudol to provide the aromatic ring of a protoilludane as shown. With the cyclobutane ring fused to the cyclohexane, there is a quaternary methylated center which cannot easily undergo some kind of elimination to aromatize. Ref 21 provides an explanation but I think it would be helpful for readers of THIS paper to see how this may happen, so please expand upon the cyclobutyl ring opening mechanism.
  • Line 159: Change to “This prompted us to evaluate…”
  • Line 168: Please ensure the table caption is on the same page as the table.
  • Line 177: Change to :…induced death of 100% of treated…”
  • Supplementary information file: Please add NMR field strengths to the figures. Please provide the HR-mass spectrum for compound 1. Please also provide an expansion of the region around 2.8 ppm in the 1H NMR spectrum as there are four overlapping resonances here and I would like to be able to see how they were resolved as listed in the NMR table.

Author Response

My main suggestion is the naming of the new compound. Given its high structural similarity to the existing echinolactone family, especially echinolactone B, and that echinolactone D was also isolated, I think it is a stretch to give compound 1 a whole new name. I therefore request naming compound 1 as echinolactone E.

Answer: We understand the suggestion of the reviewer; however, we would prefer to maintain the new name asperorlactone to underline the novelty of the fact that this class of compounds had never been found from Aspergillus and from ascomycetes, in general.

Line 21: The word novel is used far too often, and compound 1 is closely related to other known metabolites therefore replace “novel” with “new”. Novel should be restricted in use only to compounds with new carbon skeletons.

Answer: The word "novel" has never been used in the manuscript. We have now replaced "undescribed" with "new".

Line 24: Change to “… collected from Red Sea marine sediments.” Line 29: Remove plural from zebrafish embryo. There is only one model here, not multiple.

Answer: These corrections in the Abstract have been done

Line 36: Remove several. Several implies only a few, and not the 100’s if not 1000’s of marine natural product chemists who have contributed over the years! Line 57: Change “was” to “were” (there were multiple samples). Line 98: Place spaces between “…27.5 x 2…” to make more clear. Line 100: Remove one from end of sentence. Line 102: Change to “…two pairs of…”

Answer: Corrections done

Line 108: Please add a few atom labels to the structure shown in figure 2 to assist the reader relate to the paragraph describing the 2D-NMR analysis.

Answer: We have added atom labels in Figure 2

Lines 110 – 115: Were there any HMBC correlations detected from H2-1? If so these should be added to figure 2 and described in the text as they would add value to the discussion.

Answer: We have added the arrows from H2-1 in Figure 2. They were already described in the text.

Line 122: Change to “…experimental and the quantum mechanically…”. Line 128: Change to “… are shown in Figure 3.” Line 147: Change to “… metabolites has, until now, been found…”

Answer: Corrections done

Line 154: I was confused by the proposed biosynthesis of 1 as I could not easily rationalize the opening of illudol to provide the aromatic ring of a protoilludane as shown. With the cyclobutane ring fused to the cyclohexane, there is a quaternary methylated center which cannot easily undergo some kind of elimination to aromatize. Ref 21 provides an explanation but I think it would be helpful for readers of THIS paper to see how this may happen, so please expand upon the cyclobutyl ring opening mechanism.

Answer: We thank the reviewer for spotting this. Figure 5 has been expanded to provide further clues on the biosynthetic origin of asperorlactone

Line 159: Change to “This prompted us to evaluate…” Line 168: Please ensure the table caption is on the same page as the table. Line 177: Change to :…induced death of 100% of treated…”

Answer: Corrections done.

Supplementary information file: Please add NMR field strengths to the figures. Please provide the HR-mass spectrum for compound 1. Please also provide an expansion of the region around 2.8 ppm in the 1H NMR spectrum as there are four overlapping resonances here and I would like to be able to see how they were resolved as listed in the NMR table.

Answer: The field strength has been added to the figures. The HR-mass spectrum has been added to the Supplementary (Figure S6). The expansion has been provided, however, the table has been corrected.

Round 2

Reviewer 2 Report

The manuscript is now suitable for publication in Marine Drugs.